

# Ordinary kriging vs inverse distance weighting: spatial interpolation of the sessile community of Madagascar reef, Gulf of Mexico

Salvador Zarco-Perello[1,2] and Nuno Simões[1]

[1] Unidad Académica Sisal, Facultad de Ciencias, Universidad Nacional Autónoma de México, Sisal, Yucatán, México
[2] School of Biological Sciences and UWA Oceans Institute, University of Western Australia, Perth, Western Australia, Australia

Corresponding author
Salvador Zarco-Perello,
salvador.zarco.perello@gmail.com

## ABSTRACT

Information about the distribution and abundance of the habitat-forming sessile organisms in marine ecosystems is of great importance for conservation and natural resource managers. Spatial interpolation methodologies can be useful to generate this information from *in situ* sampling points, especially in circumstances where remote sensing methodologies cannot be applied due to small-scale spatial variability of the natural communities and low light penetration in the water column. Interpolation methods are widely used in environmental sciences; however, published studies using these methodologies in coral reef science are scarce. We compared the accuracy of the two most commonly used interpolation methods in all disciplines, inverse distance weighting (IDW) and ordinary kriging (OK), to predict the distribution and abundance of hard corals, octocorals, macroalgae, sponges and zoantharians and identify hotspots of these habitat-forming organisms using data sampled at three different spatial scales (5, 10 and 20 m) in Madagascar reef, Gulf of Mexico. The deeper sandy environments of the leeward and windward regions of Madagascar reef were dominated by macroalgae and seconded by octocorals. However, the shallow rocky environments of the reef crest had the highest richness of habitat-forming groups of organisms; here, we registered high abundances of octocorals and macroalgae, with sponges, *Millepora alcicornis* and zoantharians dominating in some patches, creating high levels of habitat heterogeneity. IDW and OK generated similar maps of distribution for all the taxa; however, cross-validation tests showed that IDW outperformed OK in the prediction of their abundances. When the sampling distance was at 20 m, both interpolation techniques performed poorly, but as the sampling was done at shorter distances prediction accuracies increased, especially for IDW. OK had higher mean prediction errors and failed to correctly interpolate the highest abundance values measured *in situ*, except for macroalgae, whereas IDW had lower mean prediction errors and high correlations between predicted and measured values in all cases when sampling was every 5 m. The accurate spatial interpolations created using IDW allowed us to see the spatial variability of each taxa at a biological and spatial resolution that remote sensing would not have been able to produce. Our study sets the basis for further research projects and conservation management in Madagascar reef and encourages similar studies in the region and other parts of the world where remote sensing technologies are not suitable for use.

## INTRODUCTION

Coral reefs are important centres of biodiversity (*Plaisance et al., 2011*) that provide multiple natural resources and ecosystem services to human societies (*Mumby et al., 2011*). However, multiple disturbances are impacting these ecosystems and causing changes in their community structure (*Norström et al., 2009*). The main groups of sessile organisms inhabiting coral reefs are hard corals (Scleractinia and millepores), macroalgae, octocorals, sponges, and zoantharians (*Lewis, 2006*; *Diaz & Rützler, 2001*; *Norström et al., 2009*; *Wee et al., 2017*); these taxa are habitat-forming organisms (HFO) that shelter many species of fishes, echinoderms, gastropods, mollusks and crustaceans (*Duffy, 1992*; *Goh, Ng & Chou, 1999*; *Pérez, Vila-Nova & Santos, 2005*; *Santavy et al., 2013*; *Cházaro-Olivera & Vázquez-López, 2014*), which in turn sustain the fisheries and tourism industries of the world (*Moberg & Folke, 1999*). Hard corals used to dominate the seascape of tropical reefs; however, their populations have declined in recent decades due to multiple disturbances such as overfishing, eutrophication and high temperatures (*Nyström & Folke, 2001*), allowing other HFO to increase their abundance (*Nyström, Folke & Moberg, 2000*; *Wilkinson, 2004*; *Ruzicka et al., 2013*; *McMurray, Finelli & Pawlik, 2015*). Many coral reefs are now dominated by macroalgae (*McManus & Polsenberg, 2004*), with other reefs presenting high percentages of substrate covered by octocorals, zoantharians and sponges (*Norström et al., 2009*; *Cruz et al., 2015*; *Bell et al., 2013*). Although there is uncertainty in the scientific community about the specific ecological changes that may take place in the future, it is very likely that changes will continue as global warming intensifies (*Bell et al., 2013*; *Gross, 2013*; *Ruzicka et al., 2013*). Most ecological studies have lacked a community perspective and have focused on just a few taxonomic groups, mainly Scleractinian corals and macroalgae (*McManus & Polsenberg, 2004*). However, as the community structure of coral reefs transitions, the need to monitor the distribution and abundance of all HFO has increased (*Norström et al., 2009*; *Bell et al., 2013*; *Ruzicka et al., 2013*; *McMurray, Finelli & Pawlik, 2015*).

The assessment of the abundance of all HFO is important *per se*; however, these data needs to be integrated into geographic information systems (GIS) for scientific, conservation and resource management organizations, since this allows the planning of monitoring programs and establishment of conservation areas (*Franklin et al., 2003*; *Lee et al., 2015*). Much research about the spatial distribution of the sessile benthic communities in reef ecosystems has focused on remote sensing (*Kachelriess et al., 2014*). Remote sensing technologies allow the assessment of the distribution of marine sessile organisms in extensive areas following complex procedures for atmospheric correction and spectral unmixing to achieve valid habitat classifications (*Hochberg & Atkinson, 2003*). However, accuracy of remote sensing diminishes as water turbidity and depth increase because of the light absorption by the water column (*Lucas & Goodman, 2014*). The ability of remote technologies to identify different taxa and accurately estimate their abundance

is limited (*Kutser & Jupp, 2006*) and the coarse spatial resolution of the images may not match the natural patchy variation of the sessile communities (*Andrefouet et al., 2003*; *Kachelriess et al., 2014*), requiring an *in situ* verification of the remote sensing estimations (ground-truthing), which ultimately add extra costs and effort to the studies (*Botha et al., 2013*; *Lucas & Goodman, 2014*). By now these procedures are only ideal for macro-scale studies of reefs located in clear and shallow water environments (e.g., *Zapata-Ramírez et al., 2013*). However, there are many relatively small coral reefs (e.g.,~1 km²) in deep or turbid environments which are of conservation priority (*Cohen & Foale, 2013*). In these cases, species distributions and abundance estimated through spatial interpolations based entirely on data gathered *in situ* may be more appropriate, since they do not have depth, water clarity or spatial scale limitations and can be integrated into GIS (*McClanahan, Maina & Muthiga, 2011*; *Walker et al., 2012*; *D'Antonio, Gilliam & Walker, 2016*).

There are many spatial interpolation methodologies used to predict the distribution of variables of interest in different disciplines (*Li & Heap, 2008*). Among them, kriging, a geostatistical methodology, and inverse distance weighting (IDW), a simpler non-geostatistical methodology, have been used widely to predict many environmental and agricultural variables (reviewed by *Li & Heap, 2008*; *Li & Heap, 2011*; *Li & Heap, 2014*), such as soil fertility (*Mueller et al., 2004*), mud content (*Li et al., 2011*) and bathymetry (*Bello-Pineda & Hernández-Stefanoni, 2007*). Published studies in marine ecology applying these methods to generate distribution and abundance maps of marine organisms are less common. Kriging has been used for crustaceans (*Rufino et al., 2005*; *Surette, Marcotte & Wade, 2007*), echinoderms (*Hernandez-Flores et al., 2015*), fish (*Rueda & Defeo, 2003*; *De Mazières & Comley, 2008*; *Ruppert et al., 2009*), seagrass (*Holmes et al., 2007*), hard corals and encrusting algae (*Knudby et al., 2013*); whereas studies using IDW have been limited to mollusks (*Berry, Hill & Walker, 2016*), coral cover (*Walker et al., 2012*; *D'Antonio, Gilliam & Walker, 2016*), coral diameters (*Burman, Aronson & Van Woesik, 2012*) and coral and fish diversity (*McClanahan, Maina & Muthiga, 2011*). However, no study has applied these two methodologies across different HFO.

The Gulf of Mexico has many important reef systems that have been studied extensively (*Chávez, Tunnell Jr & Withers, 2007*; *Hickerson et al., 2008*; *Horta-Puga et al., 2015*). Nonetheless, there exists a dearth of information about many other smaller reefs located on the Yucatan continental shelf and developing in turbid waters, despite being important centres of biodiversity and fishery resources (*Zarco-Perello et al., 2013*). Our study (i) gathered baseline information of the abundance and community structure of all HFO inhabiting one of these poorly studied reefs and (ii) compared the accuracy of IDW and OK to interpolate their abundances with data sampled at different distances and synthesize this information into a map of HFO richness.

## MATERIALS AND METHODS

### Sampling design

From August to October of 2007 we completed 15 photo-transects across the extent of Madagascar reef in the Gulf of Mexico; each transect measured 200 m in length and

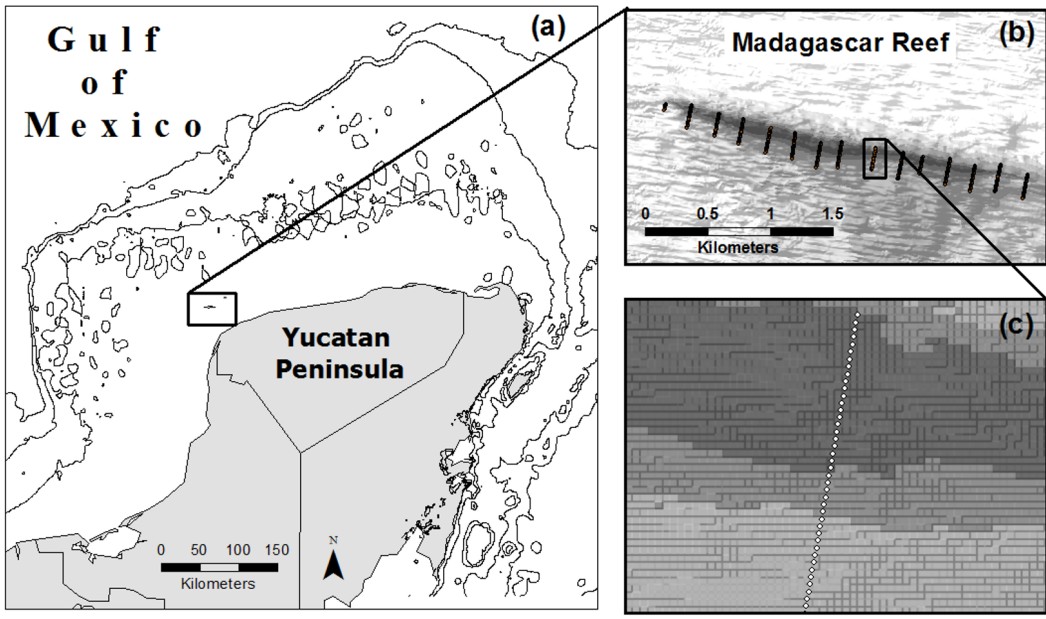

**Figure 1** Location of Madagascar reef in the Gulf of Mexico (A) and distribution of sampling points across the bathymetric gradient along the reef (B, C).

was located 200 m from adjacent transects (*Zarco-Perello, Moreno-Mendoza & Simões, 2014*) (Fig. 1). Photo-quadrats (0.8 m$^2$) of the benthos were taken every 5 m along the transects, each one representing a sampling point ($n = 580$). We recorded information about geographic coordinates, depth, substrate type (i.e., rock, sand and rubble) and reef region (i.e., windward, crest and leeward) of each sampling point.

## Community structure analyses

Relative abundance of all the HFO was estimated as percent cover in each photograph using the point-count method (see *Fabricius & McCorry, 2006*; *Ruzicka et al., 2013*). Biological similarities between the different regions of the reef were analysed with non-metric Multi-Dimensional Scaling (nmMDS) and an Analysis of Similarity (ANOSIM) to test for statistical significance; data was log transformed and the distance matrix was calculated with the bray-curtis method using the software R v.3.1.3 (vegan package) (The R Foundation) with the interphase RStudio v.0.99.473 (Rstudio, Inc.).

## Spatial interpolation analyses

The spatially referenced percent cover data from each photograph was used to interpolate the abundance of each HFO using IDW and Ordinary Kriging (OK), the most recommended univariate method of kriging (*Li & Heap, 2014*). IDW and OK interpolations are based on the principle of spatial autocorrelation of samples by distance, where the closer the samples are from each other, the more similar would be their values. Under this principle, the prediction of a value in an unsampled place is calculated by giving more weight to samples that are closer to the prediction point. However, IDW uses arbitrary

exponential weighting of the influence that each sample has according to distance, whereas OK involves a process of variography to model the spatial autocorrelation of the data to assign weights, which can result in better interpolations under an appropriate sampling design; nonetheless it is time consuming and it is still subjective since it involves many user decisions (*Li & Heap, 2014*). Finally, both interpolators use a determined quantity of observations for the predictions; these observations must be located within a 'searching window', an area around the point of prediction, which geometry is defined by the user based on the empirical knowledge of the phenomena under study (*Li & Heap, 2014*).

The interpolation analyses of OK and IDW were done considering different sampling distances (5, 10 and 20 m) to evaluate the effort required to capture the spatial structure of each HFO. Modelling parameters changed under each sampling distance. For OK, the models (e.g., spherical and exponential) that best fitted the data of the variograms of each HFO were selected; for IDW, different power values (i.e., 1, 2, 3) were used as weighting factors for each HFO. The parameters of the searching window (i.e., length of axis 1 and axis 2) were the same for both methodologies. The best models of OK and IDW for each HFO under each sampling distance were selected following cross-validations (*Goovaerts, 1997*; Supplemental Information). The performance of each methodology was compared based on the absolute mean error (ME) of their predictions (i.e., based on the absolute values of the errors), the regression coefficient ($r^2$) of predicted against measured values, graphical comparisons (box-plots) of the distribution of predicted and measured data and visual examination of the predicted maps (*Hernandez-Flores et al., 2015*). We completed the interpolation map of hard corals only with data of *Millepora alcicornis* Linnaeus, 1758, since the inclusion of Scleractinian corals produced overestimations on the predictions given the small colony sizes found *in situ* ($<25$ cm$^2$). For such small and scattered colonies, is better to produce point maps representing presence/absence, to avoid the creation of misleading maps.

The descriptions of the spatial patterns of all HFO were based on the interpolated abundance maps of the best performing methodology (OK vs. IDW). These interpolated maps were transformed to rasters (5 m resolution) and reclassified to presence/absence with the same resolution. These rasters were used to create a map of the HFO richness on the reef by summing all the HFO present in each cell (0 to 5 scale) using the tool Raster Calculator. All the spatial interpolation analyses were done using the software ArcMap v.10.3 with the Geostatistical and Spatial Analyst extensions (ESRI corp, Redlands, CA, USA). See *Li & Heap (2008)* for a list of alternative software for interpolation methodologies.

## RESULTS

### Community structure

The sessile community of Madagascar reef differentiated spatially between the reef crest and the windward and leeward regions. The nmMDS showed two statistically significant different clusters; one belonging to sampling units from the reef crest and another one belonging to the leeward and windward regions (ANOSIM, $p = 0.001$) (Fig. 2). The reef crest presented all HFO, whereas the windward and leeward regions were greatly dominated by macroalgae, with octocorals in a lower magnitude (Fig. 3).

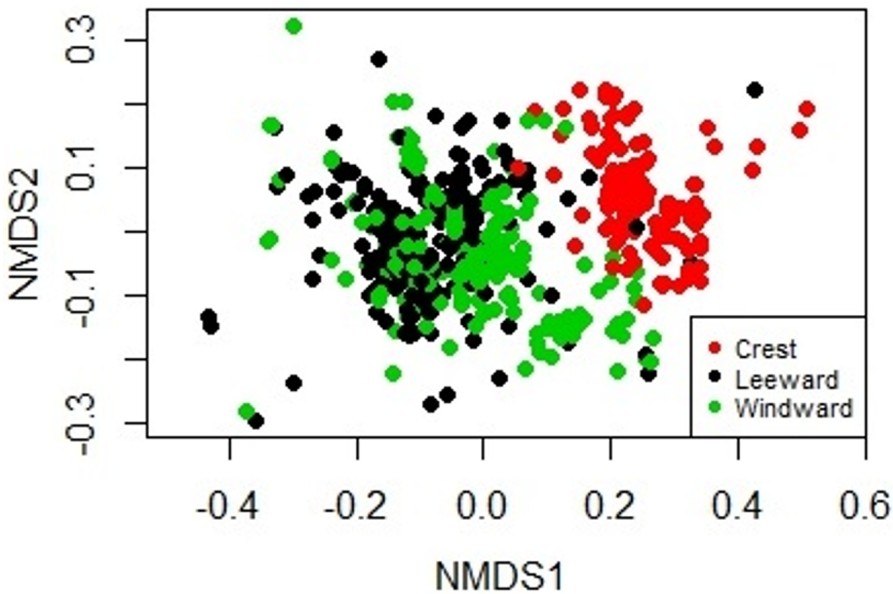

**Figure 2** Non-metric Multidimensional Scaling biplot showing the similarity on the biological composition between sampling units taken at the windward, reef crest and leeward zones of Madagascar reef, Gulf of Mexico.

The reef crest, which is the shallower and rockier region of the reef (depth: 6.8 ± 1.4 m), had the higher average abundances of hard corals (3.7%), zoantharians (*Palythoa caribeorum* Duchassaing & Michelotti, 1860 and *Zoanthus* spp.) (7.7%) and sponges (7.0%), with octocorals (44.0%) and macroalgae (37.4%) dominating the reefscape (Fig. 4). As depth increased towards the windward (depth: 16.2 ± 2.6 m) and leeward regions (depth: 14.8 ± 0.16 m), the percent cover of sandy substrate and macroalgae increased as well, whereas the abundance of all the other sessile groups decreased (Fig. 4). Macroalgae covered 80% of the substrate in each region, followed by octocorals (9% windward, 14% leeward) (Fig. 4). Hard corals, sponges and zoantharians were scarce in these environments but had slightly higher abundances at the windward (2.6%, 1.0% and 0.7%, respectively) than at the leeward region (1.4%, 0.8% and 0%, respectively) (Fig. 4). Scleractinian corals had low abundances in general but were more abundant at the windward and leeward regions (0.7%) than at the reef crest (0.1%).

## Spatial interpolations

The interpolation maps generated by both methodologies captured similar patterns of distribution and abundance for each HFO, with no anomalies detected visually. However, cross-validation analyses showed that the accuracy of the predictions changed depending on the sampling distance, the method and the particular HFO being interpolated. In general, we found that as the sampling was done at shorter distances, the predictions of both methodologies had higher accuracies. Mean errors of the interpolations were highest at the longest sampling distance (i.e., 20 m) and decreased as the sampling was

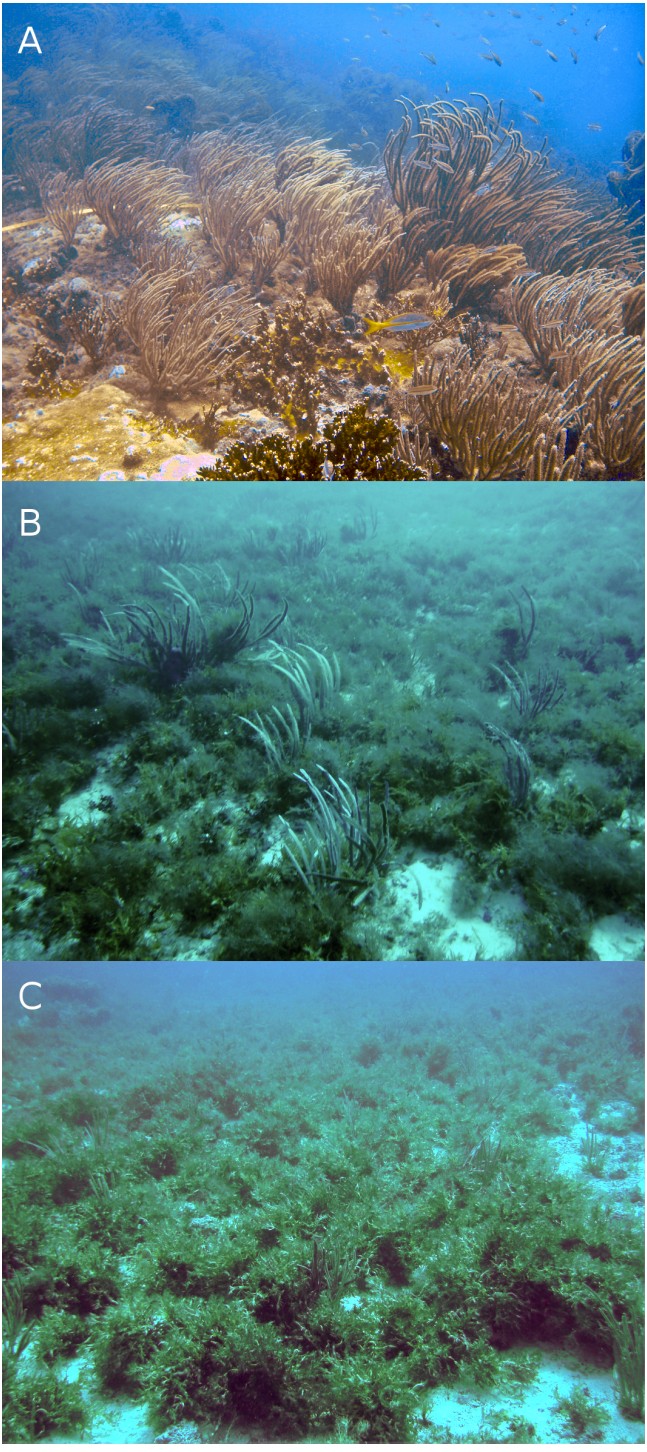

**Figure 3** Typical reefscapes of Madagascar reef, Gulf of Mexico: the shallow (depth: 6.8 ± 1.4 m) rocky reef crest (A), and the deeper sandy leeward (14.8 ± 0.16 m) (B) and windward (16.2 ± 2.6 m) (C) regions.

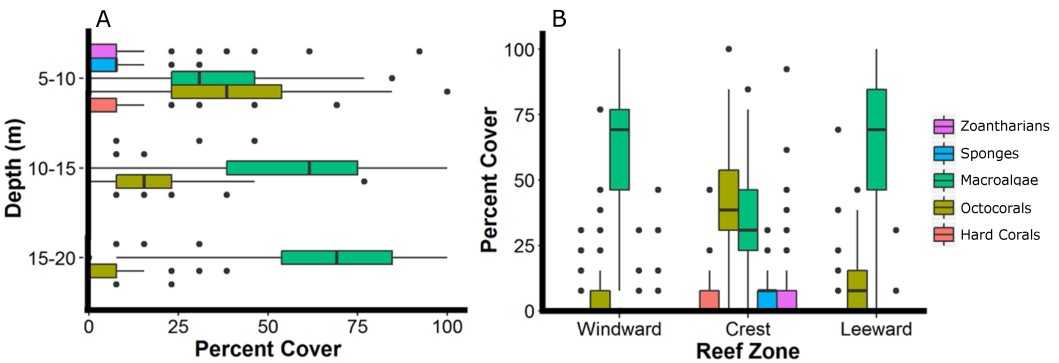

**Figure 4** **Relative abundance of the main groups of sessile organisms at different depth intervals (A) and zones (B) of Madagascar reef, Gulf of Mexico.** Box-plots represent four quartiles and median, with outliers as points.

done at shorter distances. In agreement, the correlation coefficient between measured and predicted values ($r^2$) increased as the sampling distance decreased, reaching its maximum when sampling was done every 5 m (Fig. 5). Although this pattern was generally found for both interpolation methodologies, IDW presented lower mean errors and higher $r^2$ values than OK across all HFO interpolations as the sampling was more frequent (i.e., sampling distances of 10 and 5 m); however, interpolations of both methods using data from the longest sampling distance (i.e., 20 m) had similar accuracies (Fig. 5). This pattern was more consistent considering $r^2$ rather than the mean errors of the predictions which had some contrasting values for some HFO. The interpolations with IDW of macroalgae and zoantharians had lower mean errors than OK at all sampling distances, whereas the mean errors of the interpolations of Sponges only differed clearly between methods at the shortest sampling distance (i.e., 5 m), IDW being more accurate (Fig. 5).

The distributions of the interpolated data of IDW at the shortest sampling distance, the highest accuracy, had a higher resemblance to the measured values than those generated with OK for each HFO (Fig. 6). IDW was a good predictor of the highest values of abundance measured *in situ* for all HFO, whereas the predictions of OK fell short on octocorals, sponges, zoanthids and *M. alcicornis* (Fig. 6).

The interpolation maps presented detailed information of the distribution and abundance of each HFO. The spatial distribution of all HFO was patchy, with specific areas of the reef presenting higher abundances. Macroalgae were distributed in all regions of the reef but were more abundant in deeper areas at the windward and leeward regions, where they covered up to 100% of the substrate (Fig. 7). Octocorals covered extensive areas of the reefscape as well, but patches at the central and western reef crest had higher abundances, where they covered up to 85% of the substrate (Fig. 7). Sponges were distributed all along the reef crest, presenting higher abundances (25%) at the western side, with isolated colonies in deeper zones (Fig. 7). Zoantharians were present at the reef crest, where patches at the west and east sides covered up to 45% of the substrate; their distribution was interrupted at a region where macroalgae and octocorals had high

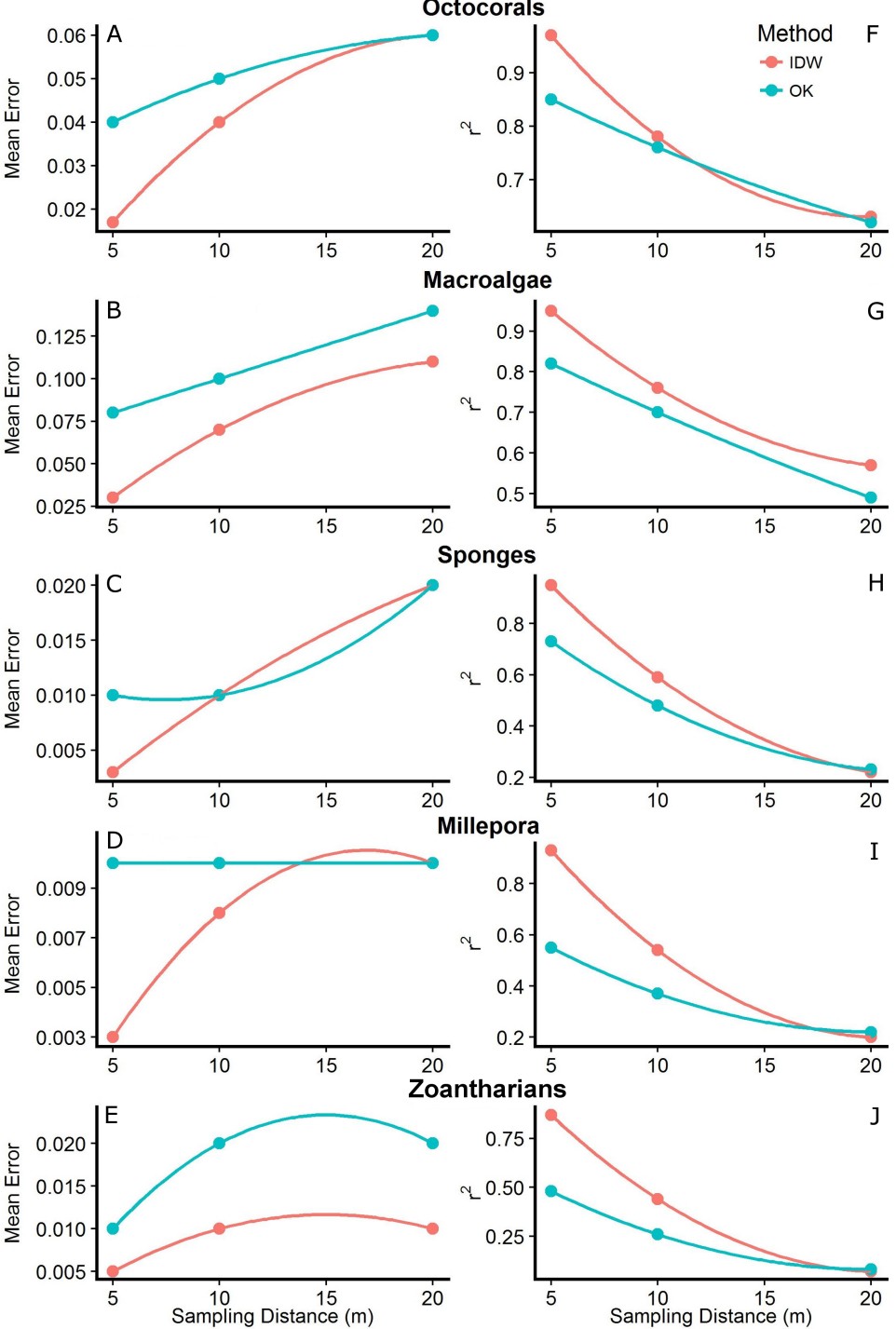

**Figure 5** Mean errors of the predictions (A–E) and correlation coefficients ($r^2$; F–J) between measured and predicted values of the interpolations from Inverse Distance Weighting (IDW) and Ordinary Kriging (OK) using datasets with different sampling distances (5, 10 and 20 m) of the abundance of macroalgae, octocorals, sponges, zoantharians and *Millepora alcicornis* (millepora) at Madagascar reef, Gulf of Mexico.

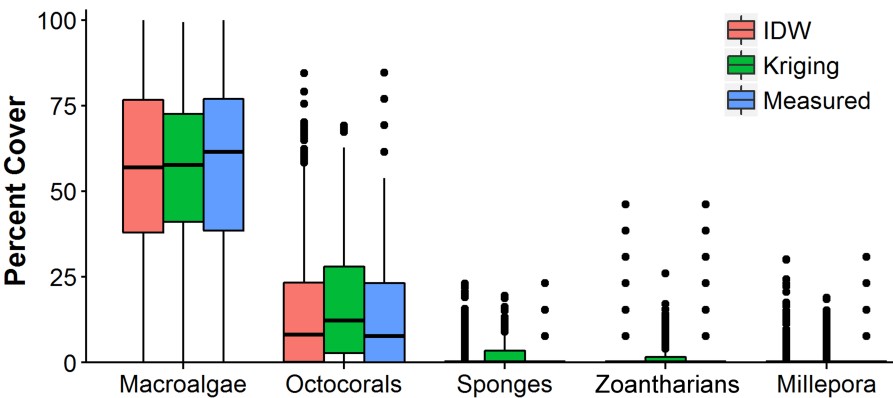

**Figure 6 Comparison of the abundance values measured *in situ* of macroalgae, octocorals, sponges, zoantharians and *Millepora alcicornis* (millepora) of Madagascar reef, Gulf of Mexico, and those interpolated by Inverse Distance Weighting (IDW) and Ordinary Kriging during cross-validation.** Box-plots represent four quartiles and median, with outliers as points.

abundances (Fig. 7). *M. alcicornis* covered the smallest area of the reef among all the HFO; colonies were distributed in three disconnected areas of the reef crest, at the western, centre and eastern regions, each presenting patches covering up to 30% of the substrate (Fig. 7); the space gaps between these areas of *M. alcicornis* had high abundances of other HFO (Fig. 7).

HFO richness had a positive relationship with depth (Fig. 8). Deep sandy areas at the windward and leeward regions had values of 1, as only macroalgae were present at these areas; slightly shallower areas, where octocorals were more common, had values of 2. Only the reef crest had extensive areas with values of 3 and 4, where either *M. alcicornis*, sponges or zoantharians were present in addition to macroalgae and octocorals. Three areas with the highest richness levels were localized at the shallowest regions of the reef crest, one at the east and two on the west side (Fig. 8).

## DISCUSSION

### Community structure

Madagascar reef sessile community differs from other reefs of the Gulf of Mexico. The windward and leeward regions of the reefs at the Campeche Bank Reef System, Veracruz Reef System and Tuxpan Reef System have been described as having important abundances of Scleractinian corals (*Chávez, Tunnell Jr & Withers, 2007*; *Larson et al., 2014*; *Horta-Puga et al., 2015*); however, at Madagascar reef, we found very small colonies and very low abundance of Scleractinian corals. Only the hard-coral *M. alcicornis* was conspicuous at the reef crest, while the leeward and windward regions were dominated by macroalgae. Our results also contrasted with reef systems further North, at the Flower Garden Banks, where surveys have reported high abundances of Scleractinian corals (>50%) and low substrate cover (<1%) of other sessile organisms, with the exception of Stetson Bank where high abundances of *M. alcicornis* (30%) and sponges (30%) have been registered (*Hickerson et al., 2008*).

# PeerJ

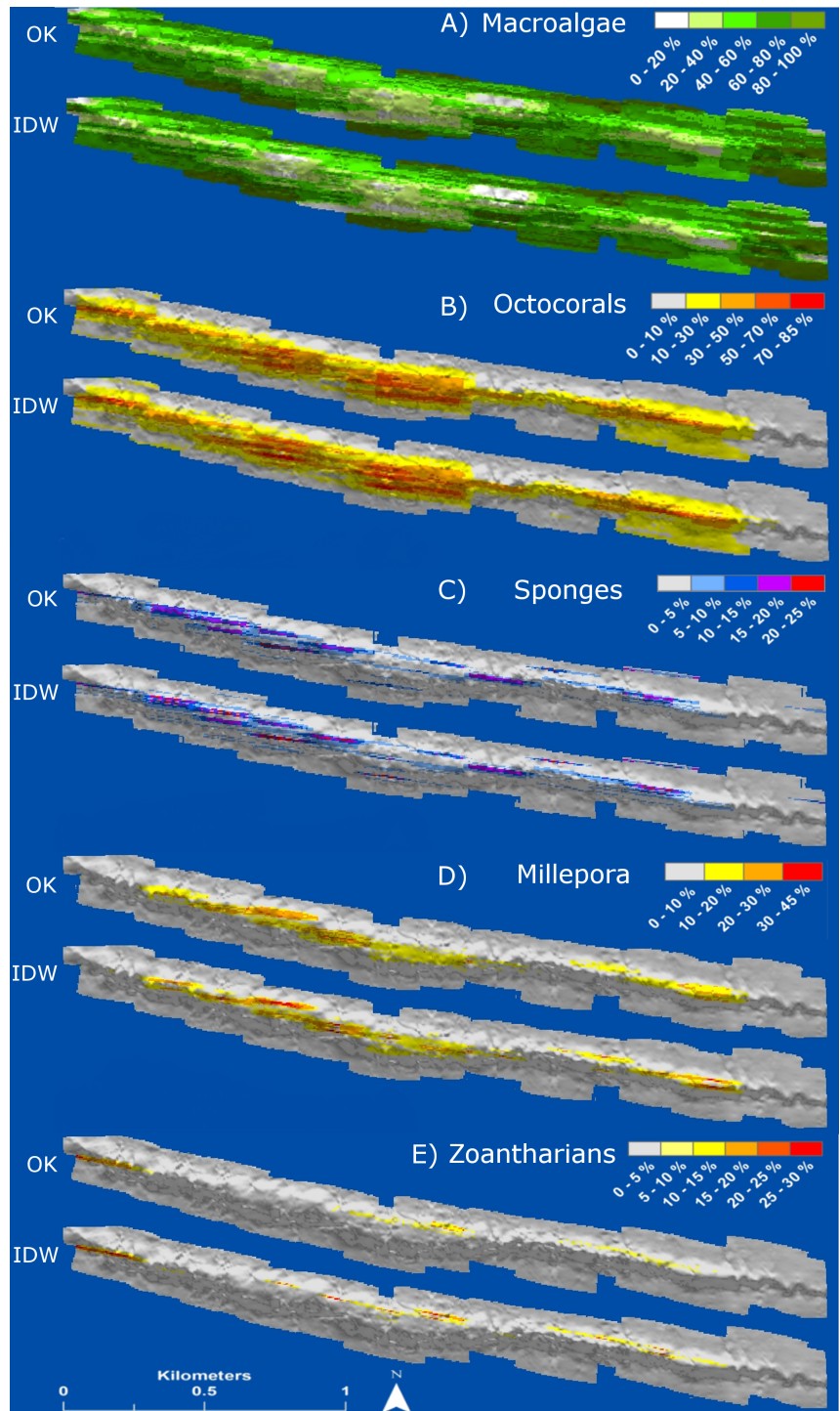

**Figure 7 Maps of distribution and abundance (percent cover) generated by ordinary kriging (OK) and inverse distance weighting (IDW) of the main groups of habitat-forming organisms of Madagascar reef, Gulf of Mexico: (A) macroalgae, (B) octocorals, (C) sponges, (D) millepora (*Millepora alcicornis*) and (E) zoantharians.**

Zarco-Perello and Simões (2017), *PeerJ*, DOI 10.7717/peerj.4078    11/24

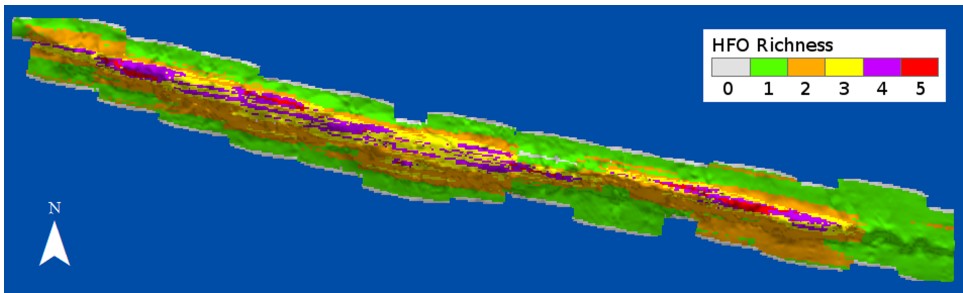

**Figure 8** **Richness of habitat-forming organisms (HFO) in Madagascar reef, Gulf of Mexico.** The calculation considers the coexistence of macroalgae, octocorals, sponges, zoantharians and *Millepora alcicornis*.

Madagascar reef crest was dominated by octocorals and macroalgae, but sponges and zoantharians were very conspicuous in different patches of the reef. Studies in the Caribbean have reported abundances of sponges as high as 24% in shallow environments (*Diaz & Rützler, 2001*), we found a lower average abundance at the reef crest (7%); however, substrate cover reached 25% in some areas. Millepores usually cover <10% of substrate over entire reefs but can be abundant in localized regions (*Lewis, 2006*), *M. alcicornis* covered ∼3% on average in Madagascar reef but in its more important areas of distribution its abundance reached 30%. Zoantharians can have high abundances in shallow environments: reefs in Brazil had ∼6% substrate covered on average (*Silva et al., 2015*) with 25% in localized areas (*Francini-Filho et al., 2013*), in St. Croix reefs had 36% cover (*Suchanek & Green, 1981*) and intertidal flats of the Southern Caribbean can cover up to half of the substrate (*Belford & Phillip, 2012*; *Rabelo et al., 2015*). Our data corresponded well with these prior studies, with zoantharians estimated to average 7.7% coverage and up to 45% coverage in core ranges.

Octocorals are a common element in reefs of the Gulf of Mexico, Caribbean (*Jordan-Dahlgren, 2002*) and other coral reefs of the world (*Fabricius & McCorry, 2006*). However, octocoral abundance can vary widely across their habitat range. Studies in the Gulf of Mexico and Caribbean have reported low abundances (2%) in Cuba (*Chiappone et al., 2001*), moderate abundances (∼16%) in the Florida Keys (*Ruzicka et al., 2013*), and high abundances (54%) across the Enmedio reef in Veracruz, Mexico (*Nelson, Stinnett & Tunnell, 1988*). Studies of octocorals outside the Latinamerican region also report this variation, with some reefs at the Great Barrier Reef in Australia showing an average of 20% cover (*Fabricius, 1997*), while high abundances have been reported in New Guinea (40%) (*Tursch & Tursch, 1982*) and the Red Sea (50%) (*Benayahu & Loya, 1981*). Octocoral abundance in Madagascar reef was similar to the highest values reported regionally and globally: 44% on average with 85% substrate covered on extensive areas of the reef crest. On the other hand, the high cover of macroalgae on all environments of the reef is not extraordinary given that this group has become dominant in many coral reef regions of the world (*Arias-González et al., 2017*).

## Spatial interpolations

Literature points out that interpolation methods are affected distinctly by some factors, such as the sampling distance or density of samples (*Li & Heap, 2008*). Our results agreed with this; we found that the accuracy of the interpolations was strongly affected by the distance between samples, especially on the interpolations of HFO that were less abundant (i.e., sponges, zoantharians and *M. alcicornis*). Interpolations based on sampling every 5 m were more accurate in comparison with sampling distances of 10 m and 20 m, which had 50% and 75% less sampling units respectably. We must consider, however, that each study requires a specific sample density, depending on the spatial variation of each phenomenon. If the variable of interest presents changes at small scales, higher sampling density will be required, such as in our case, but circumstances may arise where more spaced sampling would be appropriate (*Li & Heap, 2011*).

Sampling density affected each interpolation methodology differently. At the sampling distance of 5 m only IDW produced highly accurate interpolations for all HFO according to cross-validations, making it a very promising methodology for the interpolation of the abundance of sessile organisms inhabiting coral reefs. Although OK is generally considered a better interpolator (*Li et al., 2011*), it only generated accurate maps of macroalgae and octocorals, while the interpolations for *M. alcicornis,* sponges and zoantharians fell short of the highest values measured *in situ* as shown in the cross-validations. Nonetheless, the OK interpolations represented the distributions correctly and could allow the creation of presence/absence data, which is a common expression of abundance in ecology (*Royle & Nichols, 2003*). In contrast, IDW did not underestimate the values gathered *in situ* and displayed lower mean errors and higher $r^2$ during cross-validations. Other studies have found similar results, where kriging is not able to predict the highest values of the measured data (*Hernandez-Flores et al., 2015*) and IDW outperforms kriging interpolators. *Gong, Mattevada & O'Bryant (2014)* found that IDW was the best interpolator for arsenic concentrations in comparison with kriging, *Spokas et al. (2003)* concluded that IDW performed best for methane flux, and *Wartenberg, Uchrin & Coogan (1991)* pointed out that kriging was not superior to non-geostatistical methods at interpolating groundwater contamination despite its greater complexity.

The difference in performance between interpolation methods could be due to the variation of the data of each HFO. It has been found that variables with a high coefficient of variation (CV) are prone to have higher errors when interpolated (*Li & Heap, 2011*) and variables that have non-normal distributions are problematic to interpolate accurately with kriging since the methodology assumes a normal distribution of the variable of interest (*Heng, 2007*). This corresponds well with our results, since the abundance data of sponges, zoantharians and *M. alcicornis*, the HFO that OK failed to interpolate, had higher CV and had strongly skewed distributions in comparison to macroalgae and octocorals, which were interpolated accurately by OK. Interestingly, IDW didn't suffer in performance at the shortest sampling distance in our study regardless of the high CV of sponges, zoantharians and *M. alcicornis*.

## The importance of data spatialization

Many ecological studies in the past have assessed the abundance of different HFO, but have not presented the information in a clear spatially explicit fashion (e.g., *Newman et al., 2006*; *Ruzicka et al., 2013*, but see *Walker et al., 2012*; *Knudby et al., 2013*; *D'Antonio, Gilliam & Walker, 2016*). This is an important aspect in modern ecology that should be a standard procedure in studies regarding community structure of marine ecosystems; we show that this can be done accurately for all HFO using a simple interpolation methodology. The interpolations allowed us to see the precise spatial patterns of distribution and abundance of each HFO. Non-spatial analyses summarize ecological data and give general trends of abundance through statistical graphics (e.g., boxplots); however, without geographic coordinates, these values do not describe the spatial patterns precisely, which may obscure inferences about macro-ecological processes. Our interpolations showed how the colonizable substrate of the reef was occupied by the HFO in a mosaic fashion. Each HFO had particular areas of high abundance values at the reef crest, suggesting that despite the general dominance of octocorals and macroalgae, there is ongoing strong competition for space. All the HFO are strong competitors (*Wulff, 2006*; *Lewis, 2006*; *Rabelo, Soares & Matthews-Cascon, 2013*; *Sebens & Miles, 1988*; *Fong & Paul, 2011*; *Cruz et al., 2015*) and can influence the distribution and abundance of other groups by means of physical and chemical mechanisms that alter individual colonies and demographic processes of whole populations (*Chadwik & Morrow, 2011*).

The higher HFO richness and competition among the sessile groups at the reef crest seems to be related to depth and substrate type. The rocky substrate of the reef crest allows the recruitment of individuals from all the HFO (*Kinzie, 1973*) and the expansion of established colonies through asexual reproduction (*Jackson, 1977*). In contrast, the sandy substrate of the windward and leeward regions is unstable and primarily favours the colonization of macroalgae, likely due to their high propagation capacities through spores, faster growth rates and substrate attachment through rhizoids which anchor the organism to the sand (*Zakaria et al., 2006*; *Fong & Paul, 2011*). Larvae from other HFO might suffer high mortality rates due to sand smothering and abrasion, possibly enhanced by the high abundances of macroalgae (*Birrell, McCook & Willis, 2005*). Additionally, the reef crest is associated with higher light irradiance and water movement that can benefit all HFO, since all groups have photosynthetic species and all invertebrate HFO are suspension feeders (*Rützler, 1990*; *Lewis, 2006*; *Fabricius & De'ath, 2008*; *Fong & Paul, 2011*; *Rabelo et al., 2014*). However, octocorals are well known colonizers of turbulent environments due to their flexibility (*Sánchez, Díaz & Zea, 1997*), and their branching morphologies can give them advantage to overshadow other HFO and feed on plankton under high water flows (*Labarbera, 1984*; *Sebens, 1984*; *McFadden, 1986*; *Sebens & Johnson, 1991*; *Fabricius, Genin & Benayahu, 1995*), which could explain their higher abundances at the reef crest. Madagascar reef receives waters from an upwelling in the eastern corner of the Yucatan Peninsula (*Merino, 1997*; *Zavala-Hidalgo et al., 2006*) which supplies the nutrients to support high abundances of plankton in the region (*Ghinaglia, Herrera-Silveira & Comin, 2004*). High levels of nutrients can affect coral health (*Vega Thurber et al., 2014*) and species diversity (*Duprey, Yasuhara & Baker, 2016*), while facilitating the growth of other HFO

(*De'Ath & Fabricius, 2010*) and the establishment of sessile communities with low Scleractinian coral cover (*Arias-González et al., 2017*), such as the trend we observed in Madagascar reef.

Regions displaying the highest levels of HFO richness provide habitat heterogeneity that could benefit many mobile species. Since each of the HFO provides unique habitats where different species find refuge and food, these regions can be important centres of biodiversity in the ecosystem (*Santavy et al., 2013*). For instance, the lobster *Panulirus argus* (*Marx & Herrnkind, 1985*; *Herrnkind et al., 1997*) and the grouper *Epinephelus striatus* (*Dahlgren & Eggleston, 2000*) take refuge in sponges, octocorals and macroalgae during their juvenile stage. Many species of invertebrates (e.g., amphipods, copepods, mollusks, echinoderms and polychaetes) and fish inhabit the micro-habitats of macroalgae (*Dulvy et al., 2002*), octocorals (*Goh, Ng & Chou, 1999*), sponges (*Duffy, 1992*; *Wulff, 2006*), zoantharians (*Pérez, Vila-Nova & Santos, 2005*) and millepores (*Lewis, 2006*). Furthermore, HFO serve as food resources; Hawksbill turtles feed on sponges and macroalgae (*Bjorndal, 1990*) and several species of fish feed on zoantharians (*Francini-Filho & Moura, 2010*), macroalgae (*Choat, Clements & Robbins, 2002*) and sponges (*Pawlik et al., 1995*), while species of molluscs, echinoderms and crustaceans consume sponges as well (*Wulff, 2006*).

## CONCLUSION

The generation of spatial information about the abundance of biological organisms is needed to establish monitoring programs, detect changes in the community over time and allow conservation planning for natural ecosystems. The comparison between IDW and OK, a popular but more complex and time-consuming methodology, allowed us to conclude that, in this case, simple is best. The only published past studies using IDW in coral reef sessile organisms found this method to be a good interpolator for coral cover (*Walker et al., 2012*; *D'Antonio, Gilliam & Walker, 2016*); our results corroborate these findings and we extend the applicability of this method to other important sessile organisms that are becoming more abundant, likely because of climate change (*Norström et al., 2009*). Importantly, the sampling design, sample density and location must be adequate to the spatial variation of the organisms of interest. We showed that Madagascar reef supported important abundances of all the HFO at the reef crest region, with high HFO richness in specific areas. The accurate spatial interpolations created using IDW allowed us to see the spatial variability of each HFO at a biological and spatial resolution that remote sensing would not have been able to produce. This study provides the basis for further biological research projects and conservation management in Madagascar reef and encourages similar studies in the region and other parts of the world where remote sensing technologies are not suitable for use.

## ACKNOWLEDGEMENTS

We thank Fernando Mex and Quetzalli Hernandez for their help during fieldwork operations and Megan Sheard and all the reviewers of this work for their critic comments that helped improve this manuscript.

### Funding

This study was supported with the funding from the projects PAPIME- PE204406/PE207210, PAPIIT-IN216506-3 of UNAM and CONACyT-SEMARNAT No. 108285 granted to Nuno Simões. The funders had no role in study design, data collection and analysis, decision to publish, or preparation of the manuscript.

### Grant Disclosures

The following grant information was disclosed by the authors:
UNAM: PAPIME- PE204406/PE207210, PAPIIT-IN216506-3.
CONACyT-SEMARNAT: 108285.

### Competing Interests

The authors declare there are no competing interests.

### Author Contributions

- Salvador Zarco-Perello conceived and designed the experiments, performed the experiments, analyzed the data, contributed reagents/materials/analysis tools, wrote the paper, prepared figures and/or tables, reviewed drafts of the paper.
- Nuno Simões conceived and designed the experiments, contributed reagents/materials/analysis tools, reviewed drafts of the paper.

### Data Availability

    The raw data has been provided as Supplemental Files.

### Supplemental Information

Supplemental information for this article can be found online at http://dx.doi.org/10.7717/peerj.4078#supplemental-information.

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
