# Peer review of "Ordinary kriging vs inverse distance weighting: spatial interpolation of the sessile community of Madagascar reef, Gulf of Mexico"

_PeerJ, doi:10.7717/peerj.4078_

## Round 0.1 · original submission · Major Revisions

All three reviewers were generally positive about your work, and have offered constructive criticism; please go over their comments carefully. As well, two of the reviewers have noted the English needs work - please find a colleague to thoroughly edit the work, as PeerJ does not provide English editing services. Thus, with these two factors combined, my decision is "major revisions" are needed.

I look forward to seeing a revised version of your work.

·

Basic reporting

In my review of the document, I went through and made quite a few modifications of sentence structure and grammar. The paper is nearly there, but the overall quality of the writing would be well-served by another edit by a native English speaker.

The authors should add the complete R code used in their analyses to the supplementary files available for download.

The authors did a fine job structuring the paper and creating some useful, interesting figures. I would like to see a more explicit hypothesis (re: IDW vs. OK) retroactively suggested in the introduction of the paper.

Experimental design

The results are interesting and useful for spatial ecologists considering IDW vs. OK procedures. As the authors stated, these methods are infrequently utilized in marine studies.

As stated above, the authors need to include their code with their other materials so readers can replicate their work.

Validity of the findings

The authors did a good job outlining the impacts and ecological rationale of their findings. I would like to see the Discussion fleshed out with a paragraph added regarding why the IDW outperformed the OK methods.

The authors successfully and frequently related the findings of this study to those of relevant prior studies.

Additional comments

I enjoyed this paper and hope to see it published. I would like to explore your R code out of my own interest in these methods and curiosity about how they perform in another study region.

·

Basic reporting

The paper is very well written and strcutured, has great figures and is generally a joy to read. It makes sense and is coherent.

Experimental design

The experimental design works, and I only have one query: there are a lot of sampling points - Madagascar Reef clearly is the best sampled reef in the world! It would be useful to undertake a sensitivity analysis to find out how many points are needed to get this level of predictive capacity - this would be informative for places with fewer survey points, and also highlight how much sampling effort is required with interpolation to get good predictive capacity.

Validity of the findings

no comment, the paper is very well rounded and the findings are valid

Additional comments

Spatial interpolation methods generally are weak or unsuitable to support spatially explicit analysis in ecology and conservation. This is why i was extremely skeptical about this article, but you succeeded in convincing me that this is a great contribution. My initial reaction also points to an area of improvement: the abstract and early introduction focus on interpolation methods, but do not explain clearly enough under which circumstances they might indeed be useful. Much spatially contiguous environmental data comes from satellite data and has a given resolution, and for projects that want to spatialise survey data at smaller scales, interpolation might be useful because environmental data that could serve as predictor variables to create spatial models do not exist.
I would also recommend to discuss a little more under which circumstances interpolation would be preferred over predictive spatial models.

Reviewer 3 ·

Basic reporting

I suggest the basic reporting can be improved. It is often ambiguous and intended messages can be unclear. The English can be improved.

The literature references and background context are good. The introduction was clear and well structured for the study. However, I suggest some of the methods is more appropriate to the discussion or Introduction.

I suggest the comparisons made in the discussion between the study site and other regions can be summarised in a table.

Attention is needed to correct english in the figure legends.

The results are relevant to the aims however the structure of the manuscript can be improved with simplified listing of the objectives.

Experimental design

I believe attention is necessary to describing the methods more clearly. Overall I understand from the manuscript that the research is relevant and adresses an overlooked approach however this can be communicated more clearly.

Validity of the findings

The conclusions are clear, however can be communicated more clearly. I believe more focus should be given to the comparison of the methods for interpolation and less to the description of the community structure at Madagascar reef. Community descriptions can be detailed in supplementary information, but I suggest this is not appropriate for a research publication and is more fitting of a survey report.

It is speculated in the abstract that patchy habitats and community composition "could benefit many mobile species" but this is not supported by findings in the paper.

Additional comments

Abstract
Text although clear could be simplified to focus on the objectives. For example from the sentence starting with “we compare…” at line 20 in the Abstract , text could be modified as follows and to also follow the order of objectives outlined at the end of the introduction:

We compare the accuracy of inverse distance weighting (IDW) and ordinary kriging (OK), to predict the distribution and abundance and identify hotspots of habitat forming organisms on a poorly studied coral reef. Hard corals, octocorals, macroalgae, sponges and zoanthids of Madagascar reef, of the Yucatan continental shelf are important biodiversity and fishery resources.

Followed by a description of the habitat:
“The deeper sandy environments of the leeward and windward regions are dominated by macroalgae and octocorals. However, the shallow rocky
environments of the reef crest have the highest richness of habitat-forming organisms with high abundances of octocorals and macroalgae, and patchy dominance of sponges, Millepora alcicornis and zoanthids, creating high
habitat heterogeneity.”

Then followed by the summary results from comparing IDW and OK…


Introduction:
The manuscript provides suitable content, structure and references in the introduction. I suggest attention is given to improve the text in line with the following examples:

Line 75: can the authors identify the “complex procedures” and/or explain why they are undesirable here.

Lines 76-78: Over use of “it’s” becomes confusing. I suggest:
“However, accuracy of remote sensing diminishes as water turbidity and depth increase light absorption by the water column (Lucas & Goodman, 2014). Remote sensing also has limited ability to identify taxa and accurately estimate abundance (Kutser & Jupp, 2006).”

Lines 84-86: Use of “having” is awkward English and I suggest modification for example as:
However, there are many relatively small coral reefs (e.g. ~1 km2), in deep or turbid environments, which are a conservation priority (Cohen & Foale, 2013).

Lines 111-114:
I suggest the outline of the objectes is reworded to combine i with ii, and iii with iv. For example:
“The present study (i) gathered baseline information of the abundance and community structure of HFO at Madagascar Reef, and (ii) compared the accuracy of IDW and OK to interpolate abundances of HFO and synthesize this information”



Methodology
Line 118: the Sampling design is a little unclear. I believe the explanation can be improved. I suggest clearly stating how many transects were used and how long they were, then how many photo quadrats were taken on each transect, followed by the size of each photoquadrat.

Line 129: I suggest changing the wording to :
“Relative abundance of habitat forming organisms was estimated as percent cover in each photograph…”

Line 139:
I suggest changing “of each photography” to “from each photograph”

Line 142-143
“…are of each other,” please change to “…are to each other…”

Information at lines 143 to 149 is possible more appropriate in the introduction or discussion.

Lines 156-157: the following sentence is unclear: “The parameters of searching window were the same for both methodologies.”

Line 163: in reference to “Millepora alcicornis (millepora)” is this a taxonomic subspecies in backrets or is this to indicate further reference in the text or graphs uses “millepora”

Lines 163-165:
“…the inclusion in the model of scleractinian corals produced overestimations on the
predictions given the small colony sizes found in situ (< 25 cm2).”
This should be discussed in terms of where the methodology is appropriate, for example are small corals common in deep and turbid environments and therefore is this method appropriate for corals?

Results
Line 192: I suggest avoinding language such as “so did”. This can be unclear for the intended meaning. A suggested alternative is: “…the percent cover of sandy substrate and macroalgae increased…”

Legend of Figure 1
Change “Localization” to “Location”

Legend of Figure 6:
Correct “fistribution”

Figure 7:
It is unclear to me the term HFO “richness” is appropriate. The description of how this was calculated at line 169 in the methods indicates it is a presence absence result to show how many types of HFO are present.

Discussion
Line 303: Start sentene with a capital “however,…”

I suggest that the comparison of the community structure of Madagascar reefs is not the key result and therefore should not occupy a large initial section of the discussion. Perhaps this can be summarized in a table in the results? It is currently confusing to follow, partly as a result of the use Endlish language, the multiple locations and organisms and also because the organisms referred to are not always specified. For example Lines 307 to 310 do not specify octocorals.

Lines 318-319:
I suggest rewording similarly to:
“Our study found IDW to be a better methodology than OK for interpolating the abundance of coral reef habitat forming organisms.” However I also suggest the authors discuss the limitations for small corals, as suggested in reference for Lines 163-165

Line 343-344
I suggest changing the wording as underlined: “Our interpolations showed how the space of the reef was occupied by the HFO in a mosaic fashion.”

I suggest the discussion can be started with reference to the current statement at lines 396-399, because this is the main conclusion of the study:
“The only published past studies using IDW in coral reef sessile organisms found this method as a good interpolator for coral cover (Walker et al., 2012; D’Antonio et al., 2016); our results agree with them and we extend its applicability to other important sessile organisms that are emerging under climate change (Norström et al., 2009).”

---

## Round 0.2 · Minor Revisions

While the reviewer recommended accept, they have several small comments that need addressing. In particular, upon my own review, I agree with their comment on your use of "millepore" and you need to standardize your use of this word. As well, as someone highly interested in zoantharians, I would like to ask you to 1) identify what zoantharians are present (just to genus even), and 2) if they are Palythoa spp., please replace "zoanthid" with "zoantharian", as "zoanthid" generally means from family Zoanthidae (e.g. genus Zoanthus etc).

I look forward to receiving your revised version.

·

Basic reporting

Good basic design and implementation of the study.

Experimental design

Abstract:
This is a good summary of the study

Introduction:
Scleractinian needs to be capitalised (line 47 and elsewhere)

line 63 - just a couple - replace with few

good intro

Methods
make sense

Results
Figure 3 would be better placed in the methods, where the different habitats are introduced

Discussion
terminology of Millepora is still confused, the authors use:
Millepora alcicornis (millepora)
or just the species name or just millepora.
at first mention, the authority is missing.

I suggest using Millepora alcicornis or M alcicornis for consistency.

Validity of the findings

no comment

Additional comments

na

---

## Round 0.3 · accepted · Accept

The paper has been well revised, and is now acceptable for publication. At the proof stage or earlier, please correct:
delete comma after "alcicornis" on line 171,
change "Scleractinian" to "Scleractinia" on line 48,
ensure "Millepora" in Figures 5,6,7 is in italics.

I look forward to seeing the final version published!